

# Abiotic processes control carbon dioxide dynamics in temperate karst lakes

Mariana Vargas-Sánchez[1], Javier Alcocer[2], Eric Puche[3,4] and Salvador Sánchez-Carrillo[3]

[1] Graduate Program in Marine Sciences and Limnology, Universidad Nacional Autónoma de México (UNAM), Mexico City, Mexico
[2] Tropical Limnology Group, FES-Iztacala, Universidad Nacional Autónoma de México (UNAM), Mexico City, Mexico
[3] Biogeochemistry and Microbial Ecology Dep., Museo Nacional de Ciencias Naturales (MNCN-CSIC), Madrid, Spain
[4] Cavanilles Insitute of Biodiversity and Evolutionary Biology, Universidad de Valencia, Paterna, Valencia, Spain

Corresponding author
Salvador Sánchez-Carrillo, sanchez.carrillo@mncn.csic.es

## ABSTRACT

Inland waters are crucial in the carbon cycle, contributing significantly to the global $CO_2$ fluxes. Carbonate lakes may act as both sources and sinks of $CO_2$ depending on the interactions between the amount of dissolved inorganic carbon (DIC) inputs, lake metabolisms, and geochemical processes. It is often difficult to distinguish the dominant mechanisms driving $CO_2$ dynamics and their effects on $CO_2$ emissions. This study was undertaken in three groundwater-fed carbonate-rich lakes in central Spain (Ruidera Lakes), severely polluted with nitrates from agricultural overfertilization. Diel and seasonal (summer and winter) changes in $CO_2$ concentration ($C_{CO2}$) DIC, and $CO_2$ emissions-($F_{CO2}$)-, as well as physical and chemical variables, including primary production and phytoplanktonic chlorophyll-$a$ were measured. In addition, $\delta^{13}C$-DIC, $\delta^{13}C$-$CO_2$ in lake waters, and $\delta^{13}C$ of the sedimentary organic matter were measured seasonally to identify the primary $CO_2$ sources and processes. While the lakes were consistently $C_{CO2}$ supersaturated and $F_{CO2}$ was released to the atmosphere during both seasons, the highest $C_{CO2}$ and DIC were in summer (0.36–2.26 $\mu$mol $L^{-1}$). Our results support a strong phosphorus limitation for primary production in these lakes, which impinges on $CO_2$ dynamics. External DIC inputs to the lake waters primarily drive the $C_{CO2}$ and, therefore, the $F_{CO2}$. The $\delta^{13}C$-DIC signatures below –12‰ confirmed the primary geogenic influence on DIC. As also suggested by the high values on the calcite saturation index, the Miller-Tans plot revealed that the $CO_2$ source in the lakes was close to the signature provided by the fractionation of $\delta^{13}C$-$CO_2$ from calcite precipitation. Therefore, the main contribution behind the $C_{CO2}$ values found in these karst lakes should be attributed to the calcite precipitation process, which is temperature-dependent according to the seasonal change observed in $\delta^{13}C$-DIC values. Finally, co-precipitation of phosphate with calcite could partly explain the observed low phytoplankton production in these lakes and the impact on the contribution to increasing greenhouse gas emissions. However, as eutrophication increases and the soluble reactive phosphorus (SRP) content increases, the co-precipitation of phosphate is expected to be progressively inhibited. These thresholds must be assessed to understand how the $CO_3^{2-}$ ions drive lake co-precipitation dynamics. Carbonate regions extend over 15% of the Earth's surface but seem essential in the $CO_2$ dynamics at a global scale.

## INTRODUCTION

Inland water carbon dynamics rely on the complex interactions between the ecosystem compartments at different scales, including the water body, the watershed, and the atmosphere (*Butman et al., 2018*). During carbon (C) processing in inland waters, carbon dioxide ($CO_2$) is consumed and released through autotrophic and heterotrophic processes under aerobic and anaerobic conditions (*Ortiz-Llorente & Alvarez-Cobelas, 2012*; *Raymond et al., 2013*; *Li et al., 2021*), but also through abiotic processes related with carbonate equilibrium, which consume or add large amounts of $CO_2$ into the aquatic systems (*Liu, Dreybrodt & Wang, 2010*). Therefore, freshwater contributes significantly to the global fluxes of $CO_2$, accounting for approximately 50% of the terrestrial C sink (*Tranvik et al., 2009*; *Bastviken et al., 2011*). Lakes, in particular, play a crucial role in the C cycle, with global $CO_2$ released estimated to reach up to 1.1 Pg $CO_2$ yr$^{-1}$ (*Raymond et al., 2013*; *Deemer et al., 2016*).

$CO_2$ production, transport, and emission processes in lakes are strongly influenced by multiple internal and external factors such as hydrodynamics, pH, temperature, and the availability of organic substrates, oxygen, and nutrients (*Beaulieu, DelSontro & Downing, 2019*; *Xiao et al., 2020*). Lakes are usually oversaturated with $CO_2$ and act as a C source of $CO_2$ release into the atmosphere (*Ni, Luo & Li, 2019*). The amount of $CO_2$ dissolved in a water body and the subsequent release of $CO_2$ at the water-air interface depends on three main processes: (1) the surplus of $CO_2$ derived from the C respiration, including anaerobic oxidation, which exceeds photosynthesis uptake because dissolved and particulate organic carbon (OC) inputs are in excess (*Natchimuthu et al., 2017*) and methane production, coupled to other metabolic anoxic processes such as denitrification or sulfate reduction (*Martinez-Cruz et al., 2018*); (2) the excess of $CO_2$ entering the lake through surface or groundwater inflows derived from weathering and terrestrial organic matter respiration in the watershed (*Marcé et al., 2015*); and (3) $CO_2$ released during carbonate precipitation (*Stumm & Morgan, 1996*).

Karst regions cover approximately 15% of the Earth's geographical area (*Cao et al., 2023*), with carbonate rocks accounting for 94% of the carbon pool (*Xiong et al., 2022*), and the carbon sink effect driven by the karstification process being recognized as an important fraction of the global carbon cycle on short time scales (*Binet et al., 2022*; *Cao et al., 2023*). The global carbonate sink is equivalent to 74.5% of the global net forest sink and represents 28.7% of terrestrial sinks or 46.8% of the missing sink (*Li et al., 2018*). Thus, the $CO_2$ budget in karst lakes relies on interactions between abiotic and biotic processes (*Liu, Dreybrodt & Wang, 2010*). Several studies reveal that carbonate lakes may act as both sources and sinks of $CO_2$ depending on the signs of the interactions between lake metabolisms, geochemical processes, and groundwater inputs (*Pu et al., 2017*; *Zhang et al.,*

*2017*). Nevertheless, the complexity underlying all the $CO_2$ sources makes it very difficult to identify the main mechanisms controlling the net balance of $CO_2$ in karstic lakes.

Karst systems greatly rely on groundwater flow and geochemistry but demonstrate high sensitivity to other environmental impacts and human pressures (*Ni et al., 2021*; *Cao et al., 2023*). For example, the effects of eutrophication on $CO_2$ emissions from lakes are controversial because the effect of plankton metabolism should not be considered in isolation as the only mechanism that controls the carbon budget; contrarily, several mechanisms are triggered, including microbial processes using organic or inorganic substrates and other geochemical processes, which release or consume $CO_2$, and it is unclear when and how the C sink capacity of lakes changes to become a source (*IPCC, 2013*; *Wen et al., 2017*).

In addition to agriculture, urban development is one of the main threats faced by the lakes. Urbanization has been reported to affect the carbon biogeochemical cycle of water bodies, contributing to even higher carbon dioxide ($CO_2$) outgassing from the aquatic ecosystems to the atmosphere (*Tang, Xu & Li, 2021*; *Cao et al., 2023*). Although the drivers of ecosystem change are similar in urban and natural lakes, the magnitude of the pressures on water quality associated with these drivers is likely to differ due to stronger anthropogenic forcing in urban environments (*Teurlincx et al., 2019*). Urbanization, and hence an inadequate public health system (*i.e.,* sewage collection and treatment facility) is expected to alter the net budget of $CO_2$ (*Tang, Xu & Li, 2021*), but the magnitude and sign of the $CO_2$ fluxes due to this change need yet to be clarified. Some authors (*e.g.*, *Tanner, PJ & Eyre, 2017*; *Yoon et al., 2017*; *Kumar, Yang & Sharma, 2019*) suggested that it may depend on the ability of the lakes to enhance their photosynthesis against the degradation of organic matter. However, it also depends on how additional sources of $CO_2$ increase because the environmental conditions promote other abiotic processes. Considering that abiotic processes (carbonate weathering) dominate in karst landscapes, it is essential to improve our understanding of the interactions with biotic processes to discern how the main mechanisms driving the $CO_2$ dynamics and emissions respond.

We conducted a field study to identify the key parameters regulating $CO_2$ dynamics and the main drivers of lake $CO_2$ evasion in three temperate carbonate-rich lakes exposed to elevated nitrate pollution from groundwater discharges and to different degrees of urban development. Daily and seasonal variations in $CO_2$ concentration ($C_{CO_2}$) along the water column and $CO_2$ emission rates ($F_{CO_2}$) were studied during two consecutive differentiated seasons, winter and summer, to elucidate: (1) how the $CO_2$ concentrations in the water column and fluxes in the air-water interface change at diel and seasonal scales; (2) how the DIC provided by groundwater discharge contributes to the $CO_2$ budget compared to other $CO_2$ metabolic processes, including abiotic processes; and finally, (3) how nutrient availability and limitation promote or discourage $CO_2$ evasion from the lakes. Our ultimate goal is to contribute knowledge about the feedback effects that global change promotes on karst lakes through changes in $CO_2$ evasion. Thus, by identifying the main mechanisms driving the $CO_2$ dynamics in lakes, we will be able to provide basic information so that environmental managers can develop sound and effective environmental management practices to mitigate climate change.

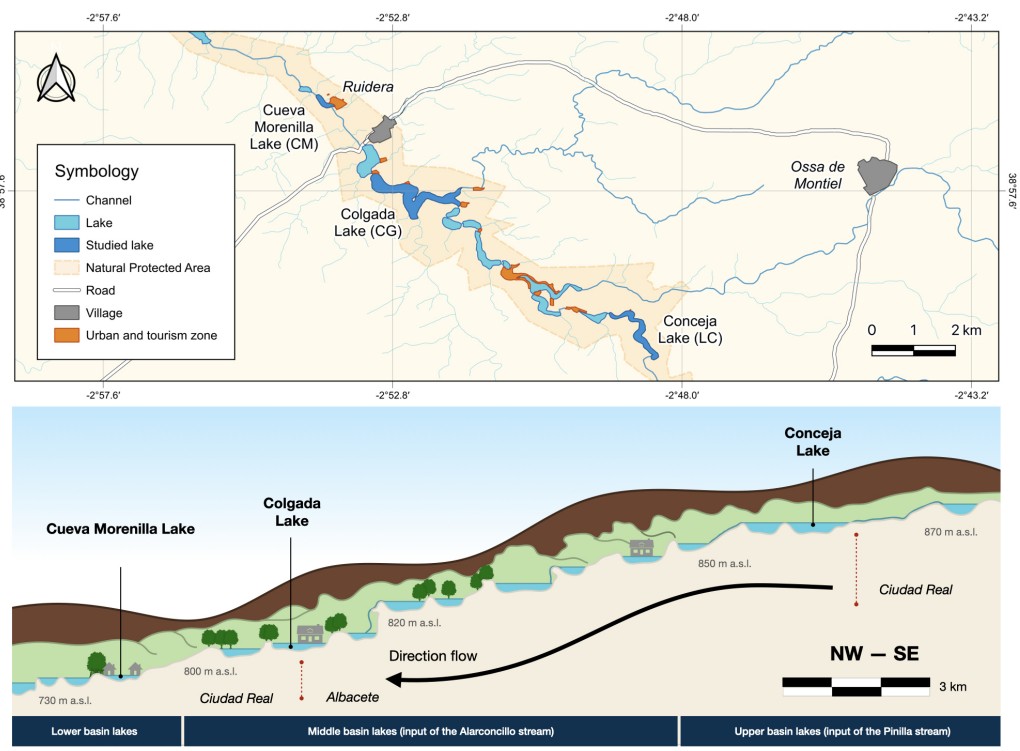

**Figure 1** Location of Lagunas de Ruidera Natural Park in Spain (top) and a conceptual illustration, showing the longitudinal profile of the lake system (bottom).

## MATERIALS & METHODS

### Study site and sampling

The Ruidera Lake district is located in central Spain (Autonomous Community of *Castilla-La Mancha*; 39°0′-38°52′N and 2°52′–2°48′E) in *La Mancha Húmeda* Biosphere Reserve (Fig. 1). Ruidera Natural Park extends over 37.72 km², comprising a chain of 15 interconnected tufa lakes developed along the watercourse of the Upper Guadiana River, with deeply karstified massive Jurassic dolostones, and dolomites, overlying Triassic sediments (*Ordóñez et al., 2005*). Lakes are mainly groundwater-fed from the Campo de Montiel aquifer, which supplies around 90% of water, while the remaining 10% comes from rainfall-runoff events (*Montero, 1994*; *Martínez-Cortina, 2001*). The aquifer is enriched with nitrate (>50 mg L$^{-1}$) from excessive agricultural fertilization used through intensive irrigation farming (*Lima, 2008*). This implies excess nitrate inputs into the lakes, mainly in those located upstream as La Conceja Lake.

The climate is subhumid temperate with dry summers (Csa, Köppen-Geiger classification), having two well-distinguished seasons: a cold/rainy winter season (from October to February) and a typical Mediterranean dry/warm summer (June to September). The average annual temperature is 15.8 °C, and the mean annual precipitation is 481 mm (AEMET Tomelloso; 39°10′9″N, 3°0′27″E, 662 m a.s.l.). Calcisols and fluvisols are
the predominant soils in the area, and the major land uses in the catchment include broad-leaved and mixed forests, meadows and pastures, and farming.

The study was undertaken in three lakes (Table S1), which were chosen as being representative of the lacustrine heterogeneity (*i.e.,* hydrological and ecological characteristics as well as anthropogenic impact—increased urbanization pressure downstream; Table S1): Conceja Lake (LC) as the most pristine groundwater-fed system located upstream of the lake chain; Colgada Lake (CG), located in the middle section of the lake systems and fed in addition by surface water and treated wastewaters from the nearby town Ossa de Montiel; and, finally, Cueva Morenilla Lake (CM), located downstream which mainly receives surface waters coming from the other upstream lakes and treated domestic wastewaters, which represent the cumulative impact of the entire lake system.

## Physical, chemical, and biological characterization

Sampling campaigns were carried out in summer (June 2021) and winter (February 2022). Monitoring consisted of three complete samplings per day for each lake in summer ($\sim$09:00 h, 15:00 h, and 21:00 h), and two samplings in winter ($\sim$09:00 h and 15:00 h), because of the shorter daylight time. Sampling collected *in situ* profiles of temperature (T, °C), dissolved oxygen (DO, mg L$^{-1}$), electrical conductivity (K$_{25}$, µS cm$^{-1}$), pH (pH units), and photosynthetic active radiation (PAR, µE m$^{-2}$s$^{-1}$) measured with a CTD (Seabird SBE 19Plus) and a YSI 6600 probe (at 1 m depth) in the maximum depth zone of the lakes. The mixing layer (Z$_{MIX}$) was estimated based on the vertical profiles of T and DO. The euphotic zone (Z$_{EU}$) was estimated from the surface down to the depth at which PAR reaches 0.1% of PAR at the surface (SPAR) (*Palmer et al., 2013*).

Water samples were taken at three depths along the vertical profile (half a meter below the surface, mid-water, and one meter above the lake bed) with a standard Niskin water sampler bottle (5 L). Water samples for nutrient analyses were placed in polyethylene bottles, stored in the dark, and frozen (−4 °C) until analysis. The nitrogen as ammonia (N-NH$_4$), nitrite (N-NO$_2$), and nitrate (N-NO$_3$), as well phosphorus as soluble reactive phosphorus (SRP), were analyzed in a segmented-flow autoanalyzer (SEAL Analytical AutoAnalyzer 3) following standardized methods (*APHA, 2005*). Water samples (60 ml) for dissolved organic carbon (DOC) were filtered through Whatman GF/F filters (0.7 µm pore size) acidified with H$_3$PO$_4$ (40%). The samples for DOC concentration were stored in amber high-density polystyrene bottles and then analyzed using Shimadzu TOC-V CSH equipment coupled with a TNM-1. The gravimetric method determined the total concentration of suspended solids (TSS, seston; *Jellison & Melack, 2001*). Chlorophyll-*a* (Chl-*a*) concentration was determined following the procedure by *Marker, Crowther & Gunn (1979)*.

Net ecosystem production (NEP) was measured at 0.5 m depth using the light-dark bottle method (*Wetzel & Likens, 2013*). Incubation lasted 4–8 h, and changes in DO concentration were measured with a HACH portable oximeter (HQ40d). Finally, carbonate precipitation was assessed through the calcite saturation index (SIc; *APHA, 2005*), an indicator of the degree of calcium carbonate saturation in water. SIc was calculated for each lake based on

T, $K_{25}$, pH values, and $HCO_3^-$ and $Ca^+$ concentrations. SIc values above 1 indicate calcite precipitation.

## Dissolved inorganic carbon and $CO_2$ flux measurements

Triplicated water samples for dissolved inorganic carbon (DIC) were collected at the three mentioned water depths along the vertical profile of the lake, ensuring no bubbles formed. An amount of 60 ml of the water samples was filtered through Whatman GF/F filters, acidified with $H_3PO_4$ (85%; pH $\approx$1), and kept cool and in the dark for 24 h, forcing the carbonate system into equilibrium (all DIC converted to $CO_2$). The headspace equilibration method (*Borges, Abril & Bouillon, 2018*) using He as carrier gas was used to extract the $CO_2$ which was then stored in 12 mL Exetainer vials (Labco Ltd., Lampeter, UK) until further analysis. Also, the headspace method, avoiding water sample acidification, was used to measure in triplicate the dissolved $CO_2$ concentration in the lake water ($C_{CO2}$; *Goldenfum, 2010*). All samples were analyzed by gas chromatography (Agilent model 6890N) equipped with a single-stage dual-packed column, where $CO_2$ was detected using a thermal conductivity detector (TCD).

Static floating chambers (5 L polyethylene), equipped with 12V electronic fans to ensure gas mixing inside the chamber, were used to measure $CO_2$ emissions and to compute total evasion fluxes ($F_{CO2}$) from the water surface (see below; *DelSontro et al., 2016*). Chamber deployment is recognized to likely enhance gas transfer through disturbance of the surface boundary layer, but the effect is mainly noted in extremely low-wind environments (*Matthews, St.Louis & Hesslein, 2003*) as well as using anchored chambers in running waters (*Lorke et al., 2015*). In Ruidera Lakes, low wind speeds are uncommon, although they can appear between 0:00 and 02:00 h (65% of cases; Sánchez-Carrillo, unpublished data). Moreover, any artificial turbulence created during gas exchange is minor for drifting (floating) chambers (*Lorke et al., 2015*). Therefore, floating chambers are considered ideal for the environment in which this study took place. Eight chambers were randomly placed in the center of each lake for 30–45 min to monitor the changes in $CO_2$ concentration inside the chamber. $CO_2$ concentration measured in the headspace of the chamber displayed a linear trend, as indicative of rate constant emission from diffusive gas flux. However, on some occasions, we observed signs of overpressure inside the chamber, when at the end of the measurements, the gas concentration described a parabola. It only occurred in five cases, when the incubations lasted for 45 min, and the procedure used to avoid these errors was to discard those final gas concentration measurements (2–3) to avoid including these data in the computation of the linear function that defined the gas emission rate during the incubations with the floating chambers. In summer, a syringe was used to extract the gas samples *via* a butyl rubber stopper in the upper chambers. The samples were stored in Exetainer vials (12 mL) until analysis by gas chromatography. In winter, a Gasera ONE PULSE based on photoacoustic spectroscopy through NDIR-PAS technology (mechanically chopped broadband IR source with optical bandpass filters) was used to measure *in situ* variations in concentration within the gas chamber. Gas measurements using the Gasera device were taken automatically at 8–9 min intervals through 2 m Teflon tubing, recirculating the measured gas back into the chambers to avoid negative pressure.

Prior intercalibration of both analytical methods (chromatography and photoacoustic spectroscopy) yielded comparable control values [$R^2 = 0.987$; [Spectros (in ppm $CO_2$) = 0.99 × chromat (in ppm $CO_2$)]].

The slope of the change in chamber $CO_2$ concentration (ppm) over the sampling time (t) was used to calculate the $F_{CO2}$. The area ($A_C$) and the volume ($V_C$) of the chamber were also considered (Eq. 1). The concentration was adjusted for pressure and temperature according to the ideal gas law.

$$\mathbf{F_{CO2}} = \frac{\mathbf{\Delta C}}{\mathbf{\Delta t}} \left( \frac{\mathbf{V_c}}{\mathbf{A_c}} \right). \tag{1}$$

## Stable isotope analyses and calculations

In each season, triplicate exetainers containing DIC (after acidification with $H_3PO_4$) and $CO_2$ (without acidification), collected as explained before, were shipped to the stable isotope lab for $\delta^{13}C$ analyses ($\delta^{13}C$-DIC and $\delta^{13}CO_2$; see below). The isotopic ratios were determined by direct injection in a Thermo Fisher Scientific Delta V Plus mass spectrometer (Thermo Fisher Scientific, Waltham, USA) connected to a GasBench system GC combustion interface ($CO_2$). These analyses were carried out in the Stable Isotope Laboratory at the University of California-Davis (SIF-UC Davis). The gas concentration in water was calculated based on the ratio between the volume of headspace and the volume of water, according to the efficiency of gas extraction (the mean extraction efficiency of DIC was 99% as determined against standard solutions of $NaCO_3$). For most gases, the heavier isotope presents a higher solubility (*Jancsó, 2002*).

Since gases with higher solubility reach equilibrium between phases quicker (*Garcia-Tigreros et al., 2016*), the rate of gas exchange for the heavier $^{13}CO_2$ isotope should be faster. Lake sediments were also analyzed to determine the $\delta^{13}C$ of organic matter ($\delta^{13}C$-OM) at the Environmental Isotope Laboratory of the University of Arizona, using a Finnigan Delta PlusXL (Thermo Fisher Scientific, Waltham, MA, USA) coupled to a Costech EA (Costech Analytical Tech Inc, Valencia, CA, USA). The isotopic data are described as:

$$\delta\left(\permil\right) = \left( \frac{R_{sample}}{R_{std} - 1} \right) \times 1000 \tag{2}$$

where R represents the isotopic ratio of the heavy isotope with the lighter isotope ($^{13}C/^{12}C$), and $R_{std}$ refers to the international standard V-PDB (fossil calcite of the Peedee formation belemnite, South Carolina), using as laboratory reference material the NIST 8545 (lithium carbonate; National Institute of Standards and Technology, Gaithersburg, MD, USA).

Carbon isotope fractionation during the precipitation of calcium carbonate was estimated, considering the fractionation of the carbon isotopes $^{12}C$ and $^{13}C$ in the equilibrium system $CO_2$ (gas)-$HCO_3^-$:

$$CaCO_3(s) + H_2CO_3(aq) \rightleftharpoons Ca^{2+}(aq) + 2HCO_3^-(aq) \tag{3}$$

$$\Downarrow$$

$$H_2O(l) + CO_2(g) \tag{4}$$

where (*s*) represents the solid-calcite, (*aq*) the aqueous solutions, and (*l*) and (*g*) the liquid and gas states of $H_2O$ and $CO_2$, respectively. Both Eqs. (3) and (4) relate to each other, in such a way that when $CO_2$ is removed from the system (*e.g.*, into the atmosphere), $H_2CO_3$ breaks apart, and the reaction shifts toward the left to replace lost bicarbonate and then calcite precipitates.

Water temperature dependence of the fractionation for $^{13}C$ between the DIC (aq) and $CO_2$ (g) was calculated according to the *Mook, Bommerson & Staverman (1974)* equation:

$$\delta^{13}C - CO_{2-fractionation}(\%) = \left(\delta^{13}C - DIC\right) + 23.644 - \left[\frac{9701.5}{(T + 273.15)}\right] \tag{5}$$

where T is the water temperature expressed in (° C), and the $\delta^{13}$C-DIC is the measured value of DIC.

The Miller-Tans *Miller & Tans (2003)* and Keeling plots *Keeling (1958)* were used to explore observed $\delta^{13}$C-$CO_2$ values and identify potential sources and fate of our lakes. Both graphical techniques are based on the principle of conservation of mass and assume mixing of isotopically distinct C sources:

$$\delta^{13}Cobs \times Cobs = \left(\delta^{13}CS \times CS\right) + \left(\delta^{13}CB \times CB\right) \tag{6}$$

where $\delta^{13}$Cobs and Cobs are the observed C isotopic composition and concentration, respectively, in each sample, resulting from a mixture of the C isotopic composition and concentration of the source ''S,'' for example, biogenic or geogenic DIC source ($\delta^{13}$CS × CS) and background "B" ($\delta^{13}$CB × CB), represented in the atmospheric $CO_2$. This principle can be expressed as a linear relationship as in Eq. (7) in terms of the Keeling plot regression and as in Eq. (8) in the Miller-Tans plot regression:

$$\delta^{13}Cobs = CB\left(\delta^{13}CB - CS\right) \times \left(\frac{1}{Cobs}\right) + \delta^{13}CS \tag{7}$$

$$\delta^{13}Cobs \times Cobs = \left(\delta^{13}CS \times Cobs\right) - \left[\delta^{13}CB(\delta^{13}CB - \delta^{13}CS)\right]. \tag{8}$$

$\delta^{13}$CS is found in the intercept of the linear regression in Eq. (4) or the slope of the linear regression in Eq. (6). The differences between the two regression models imply that the lake $\delta^{13}$C-DIC values in equilibrium with atmospheric $CO_2$ ($\delta^{13}$CB × CB) must remain fixed across observations in (Eq. 5). However, this requirement can be disregarded in Eq. (6) since $\delta^{13}$CB × CB is found in the residual variation of the regression line. The Miller-Tans plot technique is particularly suitable for approximating $\delta^{13}$Cs when including observations from multiple lakes that have undergone various degrees of $CO_2$ evasion. Both models assume linearity without further fractionation processes (*Pataki et al., 2003*). These assumptions can be violated in inland waters since kinetic fractionation processes occur when $CO_2$ is outgassed from the lake water, and other in-lake biogeochemical processes may also be involved. Accordingly, care should be taken when interpreting $\delta^{13}$CS derived from these mixing equations.

## Statistical analyses

The descriptive statistics (mean, median, and coefficient of variation) were calculated for each lake to get an overall dataset representation. The normality and homogeneity of variance of all measured variables (T, DO, pH, $K_{25}$, $Z_{EU}$, SRP, N-$NO_3$, N-$NO_2$, N-$NH_4$, TSS, Chl-a, NEP, DIC, DOC, $C_{CO2}$, and $F_{CO2}$) were examined using the Shapiro–Wilk test (*Shapiro & Wilk, 1965*) by the *shapiro.test()* function in the "stats" package (*R Core Team, 2021*). Due to the non-normal distribution of the variables, we used Mann–Whitney and Kruskal-Wallis tests to determine significant differences between the seasons (summer and winter), lakes (LC, CG, and CM), times of day (morning, evening, and night), and water sample depths (surface, middle, and bottom). The tests were performed by the functions *wilcox.test()* and *kruskal.test()* in the "stats" package (*R Core Team, 2021*). A significance level of $p < 0.05$ was used to determine the differences between the data sets. The variations among factor levels were tested using Dunn's *post-hoc* test by the function *dunnTest()* in the "FSA" package (*Ogle et al., 2023*). We then performed a principal component analysis (PCA) for each season using the "FactoMineR" package (*Lê, Josse & Husson, 2008*) to remove spurious variables not representative in the majority of DIC and $C_{CO2}$-associated processes. A redundancy analysis (RDA) using the *rda()* function from the "vegan" package (*Oksanen et al., 2022*) was also undertaken to specifically confirm the impact of environmental variables on the variability of $C_{CO2}$ for each season, but findings were statistically inconclusive and were not included in results. Finally, we conducted multiple stepwise linear regression analyses, considering T, DO, pH, $K_{25}$, TSS, Chl-*a*, nutrients, DIC, and DOC as independent variables and $C_{CO2}$ as dependent. To perform the multiple regression models (MRMs) analysis, we used the "olsrr" package's *ols_step_forward_p()* function (*Hebbali, 2020*). If necessary, the variables were transformed using a logarithmic function to meet the requirements of normality and heteroscedasticity. Finally, ordinary least squares linear regression models were performed to determine the correlations between $C_{CO2}$ and $F_{CO2}$, and in the Keeling and Miller-Tans plots. All statistical analyses and graphs were performed using RStudio (2021.09.2; *RStudio Team, 2021*).

## RESULTS

### Physical, chemical, and biological characterization

Table S2 summarizes the main physical and chemical variables of the lakes in a seasonal comparison. Although T ranged from 9.1 to 25°C and was about 12 °C higher in summer than in winter, there was no thermal stratification in any lake during the summer season. $Z_{MIX}$ and $Z_{EU}$ in all lakes were equivalent to the entire water column. Nitrogen fractions were significantly higher in lakes during summer whereas SRP showed the opposite pattern (Table 1). By far, the N-$NO_3$ was the most abundant inorganic nitrogen fraction for both seasons.

Seasonal significant differences were found in DOC, Chl-a and NEP rates, but the *post-hoc* tests revealed similar values recorded during summer and winter in the lakes (Table 2). SIc values were similar in both seasons with values ranging from 10.0–11.3 (Table 1), this being indicative of calcium carbonate ($CaCO_3$) supersaturation and probably calcite

**Table 1** Concentration (average ± standard deviation) of phosphorus as soluble reactive phosphorous (SRP), nitrogen as nitrate (N-NO$_3$), nitrite (N-NO$_2$), and ammonia (N-NH$_4$), and saturation index for calcite (SIc) in Laguna Conceja (LC), Laguna Colgada (CG), and Cueva Morenilla (CM) during summer and winter. Based on the Dunn pairwise *post-hoc* analyses after the Kruskal-Wallis tests, significant differences (at $p < 0.05$) between seasons (capital letters) and lakes (lowercase letters) were noted next to the variable values.

| Season | Lake | n | P-PO$_4$ ($\mu$mol L$^{-1}$) | N-NO$_3$ ($\mu$mol L$^{-1}$) | N-NO$_2$ ($\mu$mol L$^{-1}$) | N-NH$_4$ | SIc |
|---|---|---|---|---|---|---|---|
| Summer | | 27 | $0.1 \pm 0.04^A$ | $612.9 \pm 132.6^A$ | $3.3 \pm 0.9^A$ | $3.9 \pm 1.3^A$ | $10.44^A$ |
| | LC | 9 | $0.04 \pm 0.02^a$ | $772.6 \pm 107.2^a$ | $2.4 \pm 0.8^{ab}$ | $2.4 \pm 0.8^{ab}$ | $10.11^a$ |
| | CG | 9 | $0.06 \pm 0.03^{ab}$ | $562 \pm 25.9^{abc}$ | $3.5 \pm 0.7^{ac}$ | $5.2 \pm 0.6^c$ | $10.05^a$ |
| | CM | 9 | $0.1 \pm 0.05^{abc}$ | $504.2 \pm 9.6^{bcd}$ | $3.9 \pm 0.2^c$ | $4.1 \pm 0.5^{ac}$ | $11.17^a$ |
| Winter | | 18 | $0.8 \pm 0.5^B$ | $495.5 \pm 163.5^B$ | $1.9 \pm 0.5^B$ | $2.6 \pm 1.4^B$ | $10.69^A$ |
| | LC | 6 | $0.8 \pm 0.6^c$ | $678.5 \pm 133.4^{ab}$ | $1.9 \pm 0.3^b$ | $1.0 \pm 0.2^b$ | $10.35^a$ |
| | CG | 6 | $0.4 \pm 0.1^{bc}$ | $352.6 \pm 70.2^d$ | $1.4 \pm 0.3^b$ | $2.8 \pm 0.7^{ab}$ | $10.41^a$ |
| | CM | 6 | $1.1 \pm 0.4^c$ | $455.2 \pm 39.3^{cd}$ | $2.4 \pm 0.2^{abc}$ | $4.1 \pm 0.7^{ac}$ | $11.31^a$ |

**Table 2** Concentration (average ± standard deviation) of chlorophyll "a" (Chl-a), total suspended solids (TSS), and net ecosystem production rates (NEP) in Conceja Lake (LC), Colgada Lake (CG), and Cueva Morenilla Lake (CM) during summer and winter. Negative NEP values indicated C was lost from the lake ecosystem (NEP = GPP–R). Based on the Dunn pairwise *post-hoc* analyses after the Kruskal-Wallis tests, significant differences (at $p < 0.05$) between seasons (capital letters) and lakes (lowercase letters) were noted next to the variable values.

| Season | Lake | n | DOC mg L$^{-1}$ | Chl-a ($\mu$g L$^{-1}$) | TSS (mg L$^{-1}$) | NEP (mg C m$^{-3}$ h$^{-1}$) |
|---|---|---|---|---|---|---|
| Summer | | 27 | $1.6 \pm 0.3^A$ | $1.1 \pm 1.5^A$ | $2.1 \pm 2.2^A$ | $6.7 \pm 10.5^A$ |
| | LC | 9 | $1.4 \pm 0.2^{abc}$ | $1.1 \pm 1.5^a$ | $3.3 \pm 3.2^a$ | $17.6 \pm 6.4^a$ |
| | CG | 9 | $1.6 \pm 0.3^{ab}$ | $1.1 \pm 1.9^{ab}$ | $1.4 \pm 0.8^a$ | $-2.1 \pm 7.4^{ab}$ |
| | CM | 9 | $1.9 \pm 0.1^a$ | $0.9 \pm 0.9^a$ | $1.6 \pm 1.6^a$ | $2.0 \pm 1.1^{ab}$ |
| Winter | | 18 | $1.2 \pm 0.2^B$ | $0.3 \pm 0.1^B$ | $1.3 \pm 1.1^A$ | $-24.7 \pm 9.9^B$ |
| | LC | 6 | $1.4 \pm 0.1^{abc}$ | $0.4 \pm 0.1^{ab}$ | $1.6 \pm 1.5^a$ | $-24.8 \pm 6.8^b$ |
| | CG | 6 | $1.1 \pm 0.2^{bc}$ | $0.3 \pm 0.2^{ab}$ | $1.1 \pm 1.0^a$ | $-18.7 \pm 8.4^{ab}$ |
| | CM | 6 | $1.0 \pm 0.1^c$ | $0.2 \pm 0.1^b$ | $1.3 \pm 0.9^a$ | $-30.7 \pm 11.4^b$ |

precipitation. Only T and the SRP concentration showed temporal variation along the diel cycle (KW test, $p < 0.05$).

Among the studied lakes, LC showed the highest concentration of N-NO$_3$ (Table 1), but CM and CG had the highest N-NO$_2$ and N-NH$_4$ concentrations, respectively (Table 1). Lakes exhibited similar SRP, SIc, Chl-a, TSS, NEP, and DOC values (Tables 1 and 2). Chl-a and TSS concentrations showed vertical variation along the water column (KW test, $p < 0.05$).

## DIC and dissolved CO$_2$ concentration

The studied lakes were consistently CO$_2$ supersaturated relative to the atmospheric equilibrium concentration ($\sim$10.9 $\mu$mol L$^{-1}$) during both seasons, with higher average concentrations in summer than in winter (Fig. 2). There was no significant variation observed in $C_{CO2}$ and DIC concentrations throughout the day in any season (KW test, $p = 0.6$ for DIC and $p = 0.2$ for $C_{CO2}$). $C_{CO2}$ and DIC exhibited a weak but significant

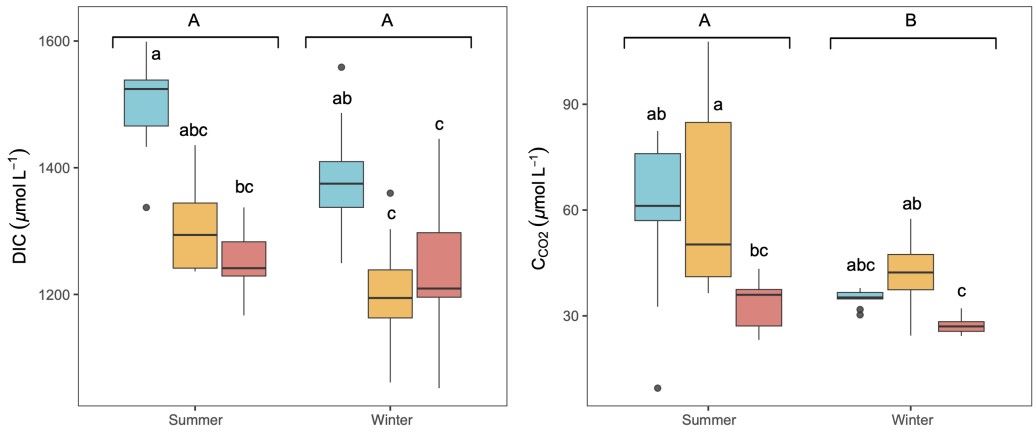

**Figure 2  Boxplot of the average dissolved CO₂ ($C_{CO2}$) and DIC concentrations along the water column during summer and winter in the three studied lakes (LC, Conceja Lake–blue; CG, Colgada Lake–yellow; and CM, Cueva Morenilla Lake–red).** The Mann–Whitney test results between seasons are indicated with brackets. Based on the Dunn pairwise *post-hoc* analyses after the Kruskal-Wallis tests, significant differences (at $p < 0.05$) between seasons (capital letters) and lakes (lowercase letters) are included. Points represent the outliers.

correlation in summer (Spearman correlation, $R^2 = 0.16$, $p < 0.05$). LC showed the highest concentration of DIC, whilst higher $C_{CO2}$ was observed in LC and CG, (Fig. 2). No significant vertical variation of DIC or $C_{CO2}$ was observed along the water column in the lakes (KW test, $p = 0.06$ for DIC and $p = 0.8$ for $C_{CO2}$).

PCAs revealed that $C_{CO2}$ was positively related to DO as opposed to photosynthesis-respiratory cycles (Fig. 3). In summer, a PCA explaining 63% of the observed variance showed DIC in the first axis together with N-NO₂, T, N-NO₃, and DOC. In winter, DIC also appeared in the first axis of the PCA, but related to $K_{25}$, and Chl-*a* (Fig. 3). The explained variance of this PCA was 60% in winter. We further identified predictors of $C_{CO2}$ by using MRM (Table 3). During the summer, $C_{CO2}$ in the water column was negatively related to T, pH, DO, and N-NO₃ which accounted for 80% of the $C_{CO2}$ variability. T was the most significant variable in the summer model (Table 3). During winter, the MRM for $C_{CO2}$ included N-NO₂, pH, and $\log_{10}$(TSS) inversely as independent variables, which together explained 70% of the $C_{CO2}$ variability. N-NO₂ was the most significant variable in the winter model (Table 3).

## CO₂ evasion fluxes ($F_{CO2}$)

CO₂ was consistently emitted from the Ruidera Lakes into the atmosphere during the day in both seasons, with an estimated mean annual emission rate of 3.64 ±2.59 g CO₂ m$^{-2}$ d$^{-1}$. Higher average seasonal $F_{CO2}$ was observed in summer, and the *post-hoc* test confirmed this pattern at LC and CM lakes (Fig. 4). During both seasons, $F_{CO2}$ values were higher in the evening and at night compared to the morning in LC and CM in summer, and LC and CG in winter (Fig. 4). Highest $F_{CO2}$ values were measured in LC and CG during summer and winter (Fig. 4).

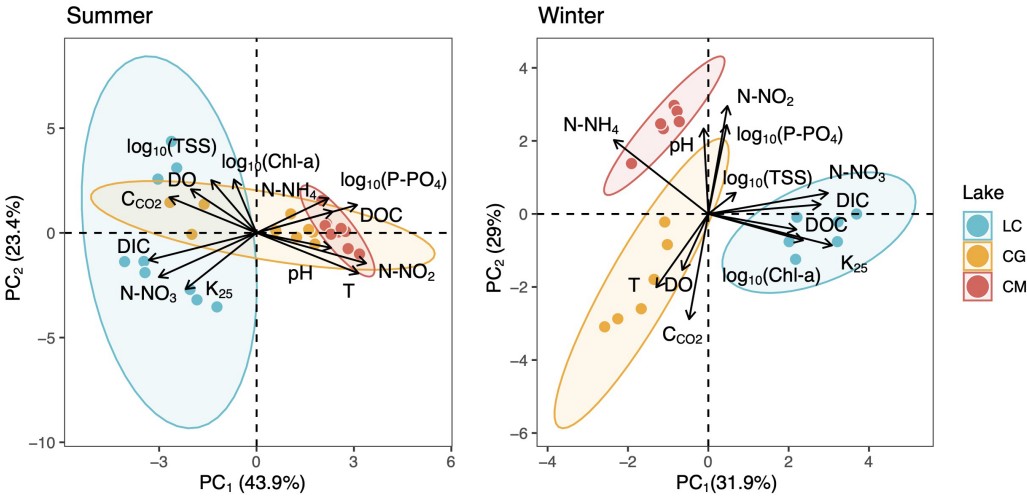

**Figure 3** Two-dimensional representation of the PCA for all measured variables during the summer **(left)** and winter **(right)**. The 95% confidence ellipses are shown for the group means. LC, Conceja Lake; CG, Colgada Lake; and CM, Cueva Morenilla Lake.

**Table 3** Multiple regression model for the $C_{CO2}$ and the environmental variables.

**Regression Summary for Dependent Variable: $C_{CO2}$**

**Summer**

| Adjusted $R^2$ = | 0.79 | F = | 18.56 | Abs. Error | 7.17 |
|---|---|---|---|---|---|
| model | Beta | Std. Error | Sig | Adj. R-Squared | R-Squared change |
| (Intercept) | 628.57 | 87.23 | 0.00 | | |
| T | −5.83 | 2.82 | 0.05 | 0.64 | |
| pH | −46.89 | 12.16 | 0.00 | 0.78 | 0.141 |
| OD | −2.28 | 1.17 | 0.07 | 0.79 | 0.004 |
| N-NO$_3$ | −0.05 | 0.02 | 0.04 | 0.79 | 0.005 |

**Winter**

| Adjusted $R^2$ = | 0.692 | F = | 13.705 | Abs. Error | 3.513 |
|---|---|---|---|---|---|
| model | Beta | Std. Error | Sig | Adj. R-Squared | R-Squared change |
| (Intercept) | 179.184 | 49.687 | 0.003 | 0 | 0 |
| N-NO$_2$ | −9.857 | 2.424 | 0.001 | 0.5724 | 0 |
| pH | −15.365 | 6.298 | 0.029 | 0.6854 | 0.113 |
| $\log_{10}$(TSS) | −5.101 | 3.148 | 0.128 | 0.7161 | 0.0307 |

Using all data, there was a positive and moderate correlation between the mean $C_{CO2}$ and $F_{CO2}$ (Pearson correlation: $F_{CO2} = 8.2 \log_{10}(C_{CO2}) - 9.8$, $p < 0.05$, $R^2 = 0.43$; Fig. 5). However, these relationships were not significant when testing seasonally (summer: $R^2 = 0.32$, $p > 0.05$, and winter: $R^2 = 0.42$, $p > 0.05$).

## $\delta^{13}$ C-CO$_2$ values and sources

There were no seasonal differences in $\delta^{13}$C-DIC, $\delta^{13}$C-CO$_2$, and $\delta^{13}$C-OM (Table 4). $\delta^{13}$CO$_2$_fract did exhibit significant seasonal differences supported by the contrasted

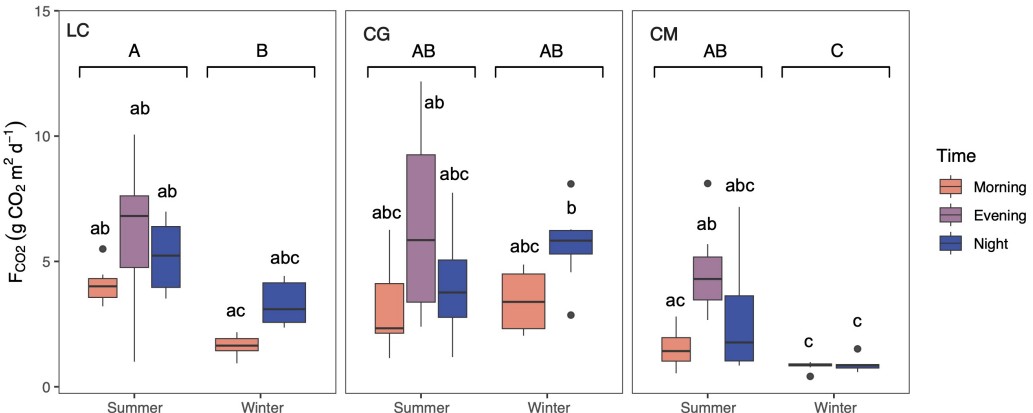

**Figure 4** **Boxplot showing the daily variation of the $CO_2$ evasion rates ($F_{CO2}$) during summer and winter in the three studied lakes (LC, Conceja Lake–left; CG, Colgada Lake–middle; and CM, Cueva Morenilla Lake–right).** The Mann–Whitney test results between seasons are indicated with brackets. Based on the Dunn pairwise *post-hoc* analyses after the Kruskal–Wallis tests, significant differences (at $p < 0.05$) between seasons (capital letters) and between lakes and times (lowercase letters) are included. Points represent the outliers.

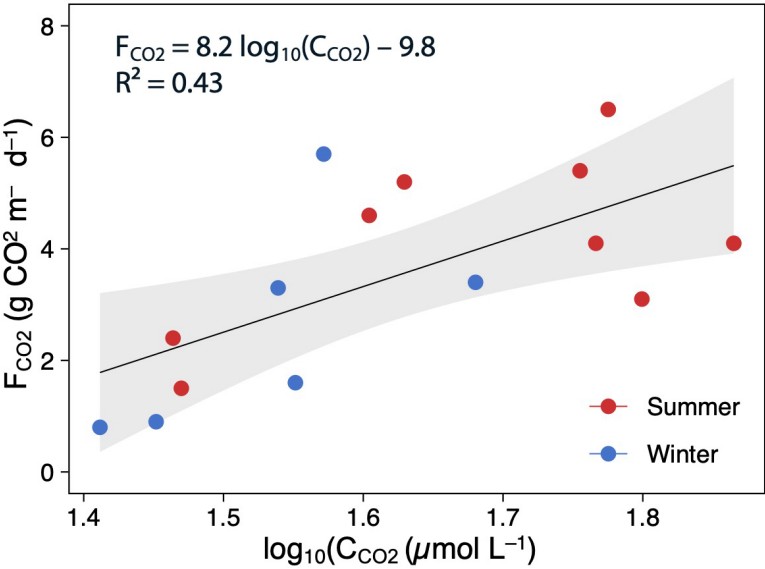

**Figure 5** **Scatterplot showing the relationships between the log($C_{CO2}$) and $F_{CO2}$.** The points are colored according to the season (summer–red; winter–blue), and the regression lines are plotted for whole (annual) data The shaded area is the 95% confidence level.

values recorded in LC-winter and CM-summer (Table 4). $\delta^{13}$C-DIC values significantly differed between LC and CM during summer (Table 4). $\delta^{13}$C-DIC and $\delta^{13}CO_2$ became enriched downstream during both seasons (Table 4). Finally, $\delta^{13}$C-OM was similar seasonally (Table 4).

**Table 4** **Summary of the average and standard deviation of $\delta^{13}$C signatures (‰) measured for DIC, $CO_2$, and sediment organic matter (OM) in Conceja Lake (LC), Colgada Lake (CG), and Cueva Morenilla Lake (CM) during summer and winter.** Data are depth-integrate averages from duplicated samples analyzed at the three depths as described in the material and methods section. $\delta^{13}$C-$CO_2$-obs are the measured values whereas $\delta^{13}$C-$CO_2$-fract are the calculated values from carbon isotope fractionation during the precipitation of calcium carbonate (Eq. 2). Based on the Dunn pairwise *post-hoc* analyses after the Kruskal-Wallis tests, significant differences (at $p < 0.05$) between seasons (capital letters) and lakes (lowercase letters) were noted next to the variable values.

| Season | Lake | $\delta^{13}$C-DIC (‰) | $\delta^{13}$C-$CO_2$ -obs (‰) | $\delta^{13}$C-$CO_2$ -fract (‰) | $\delta$13C-OM (‰) |
|---|---|---|---|---|---|
| Summer | | $-8.4 \pm 1.1^A$ | $-17.1 \pm 1.1^A$ | $-17.7 \pm 1.3^A$ | $-13.9 \pm 1.8^A$ |
| | LC | $-9.6 \pm 0.1^a$ | $-18.2 \pm 0.9^a$ | $-19.1 \pm 0.7^{ab}$ | $-12.9 \pm 0.9^{ab}$ |
| | CG | $-8.3 \pm 0.3^{ab}$ | $-16.8 \pm 2.2^a$ | $-17.6 \pm 0.8^{ab}$ | $-16.3 \pm 0.1^a$ |
| | CM | $-7.3 \pm 0.2^b$ | $-16.3 \pm 0.3^a$ | $-16.4 \pm 0.2^a$ | $-12.6 \pm 0.4^{ab}$ |
| Winter | | $-8.6 \pm 0.1^A$ | $-18.0 \pm 1.1^A$ | $-19.2 \pm 0.6^B$ | $-13.7 \pm 1.6^A$ |
| | LC | $-9.3 \pm 1.5^a$ | $-19.2 \pm 0.4^a$ | $-19.9 \pm 0.1^b$ | $-12.3 \pm 0.1^b$ |
| | CG | $-8.4 \pm 0.1^{ab}$ | $-18.1 \pm 1.1^a$ | $-19.0 \pm 0.5^{ab}$ | $-15.8 \pm 0.9^{ab}$ |
| | CM | $-8.1 \pm 0.2^{ab}$ | $-16.8 \pm 1.2^a$ | $-18.8 \pm 0.2^{ab}$ | $-13.0 \pm 0.3^{ab}$ |

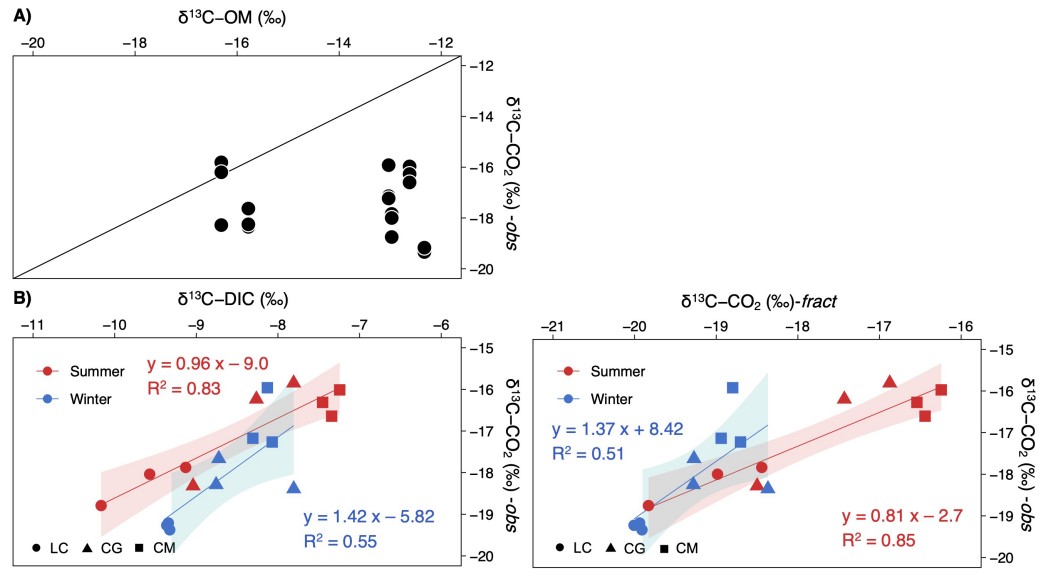

**Figure 6** **Scatterplots showing the relationships between the observed $\delta^{13}$C-$CO_2$ signatures in the studied lakes and (A) the $\delta^{13}$C of the sediment organic matter, and (B), the $\delta^{13}$C-DIC or $\delta^{13}$C-$CO_2$ calculated.** Note that each lake and season are separately represented (LC, Conceja Lake; CG, Colgada Lake; and CM, Cueva Morenilla Lake). The shaded areas are the 95% confidence level of each linear regression.

$\delta^{13}$C-$CO_2$ values were uncorrelated with the $\delta^{13}$C-OM signatures (Fig. 6A). Contrarily, $\delta^{13}$C-$CO_2$ did depend on $\delta^{13}$C-DIC, most strongly during summer (Fig. 6B). Computed signatures considering the fractionation of the carbon isotopes during calcite precipitation ($\delta^{13}$C-$CO_2$-fract) also displayed a strong correlation with observed values, mainly in summer (Fig. 6B).

The keeling plot for DIC showed a significant relationship in winter: the increase in DIC concentration led to $\delta^{13}$C-DIC enriched values; in summer, the increase in DIC concentration did not affect $\delta^{13}$C-DIC signatures (Fig. 7A). Contrarily, the increase in $C_{CO2}$ resulted in more negative $\delta^{13}$C-CO$_2$ values in both seasons, although less so in summer; however, both estimated regressions have very similar slopes for the two seasons (Fig. 7A). There were significant relationships in the Miller-Tans plot for both seasons and the whole annual data set (Fig. 7B). The linear regression models for DIC were found to be seasonally distinguishable, while those for CO$_2$ were not. The $\delta^{13}$C-DIC source values, approximated from the slope of the Miller-Tans regression models, showed strong seasonal differences, a more positive $\delta^{13}$C-DIC influence in winter ($-5.9$‰) than in summer ($-14.5$‰; Fig. 7B). The $\delta^{13}$C-CO$_2$ source values resulted very similar between seasons, consistent with the average $\delta^{13}$C-CO$_2$ calculated from the fractionation of calcite precipitation ($-18.96$‰ *vs.* $-18.47$‰, respectively; Fig. 7B). Finally, the isotope effect observed on $\delta^{13}$C-DIC values as CO$_2$ degassing downstream in the lake chain was well represented in the model depicted in Fig. 7C for the summer season; however, the effect in winter was not conclusive.

## DISCUSSION

It is widely recognized that multiple internal and external factors affect the balance of aquatic CO$_2$ (production and consumption) in inland waters (*Pu et al., 2017*; *Zhang et al., 2017*; *Li et al., 2021*). As expected, our lakes exhibited CO$_2$ supersaturation in the water column (240–856%), which acted as a CO$_2$ source to the atmosphere during both seasons. However, $C_{CO2}$ and $F_{CO2}$ were moderately dependent on each other, highlighting the importance of the physical factors such as temperature or pH controlling gas transfer at the water-air interface in these karst lakes. $C_{CO2}$ and $F_{CO2}$ showed seasonal variations, displaying an apparent strong metabolic bias associated with the warm summer. A few different processes have been cited to promote CO$_2$ supersaturation and subsequent CO$_2$ emission, including nutrient enrichment, metabolic imbalance, DOC transformations, or high allochthonous terrestrial CO$_2$ inflows (*Marcé et al., 2015*; *Weyhenmeyer et al., 2015*); however, the CO$_2$ dynamics in our carbonate lakes seems to respond strongly to other abiotic drivers such as the large amounts of DIC provided by the groundwaters and carbonate precipitation in the water column.

The increase in nutrient inputs in aquatic ecosystems alters CO$_2$ dynamics by shifting primary productivity and respiration balance (*Li et al., 2021*; *Gu et al., 2022*). Metabolism has been cited as a key driver of $C_{CO2}$ in many lakes (*DelSontro, Beaulieu & Downing, 2018*; *Ni et al., 2021*; *Li et al., 2021*). The observed change in inorganic nitrogen and phosphorus concentrations in Ruidera Lakes during the summer months was hardly associated with $C_{CO2}$, unlike those previous findings (*Wang et al., 2017*; *Wen et al., 2017*). Whereas in CM, the significant growth of phytoplankton (high Chl-*a* concentration) during summer could be leading to a reduced amount of $C_{CO2}$ in the water, as also described by *DelSontro, Beaulieu & Downing (2018)*, in LC and CG, Chl-*a* did not change; the $C_{CO2}$ pattern was unaltered, despite NEP rates at LC in summer exhibiting autotrophy (GPP > R, NEP >0). Since SRP decreased in summer, only the rise in water temperature likely enhanced

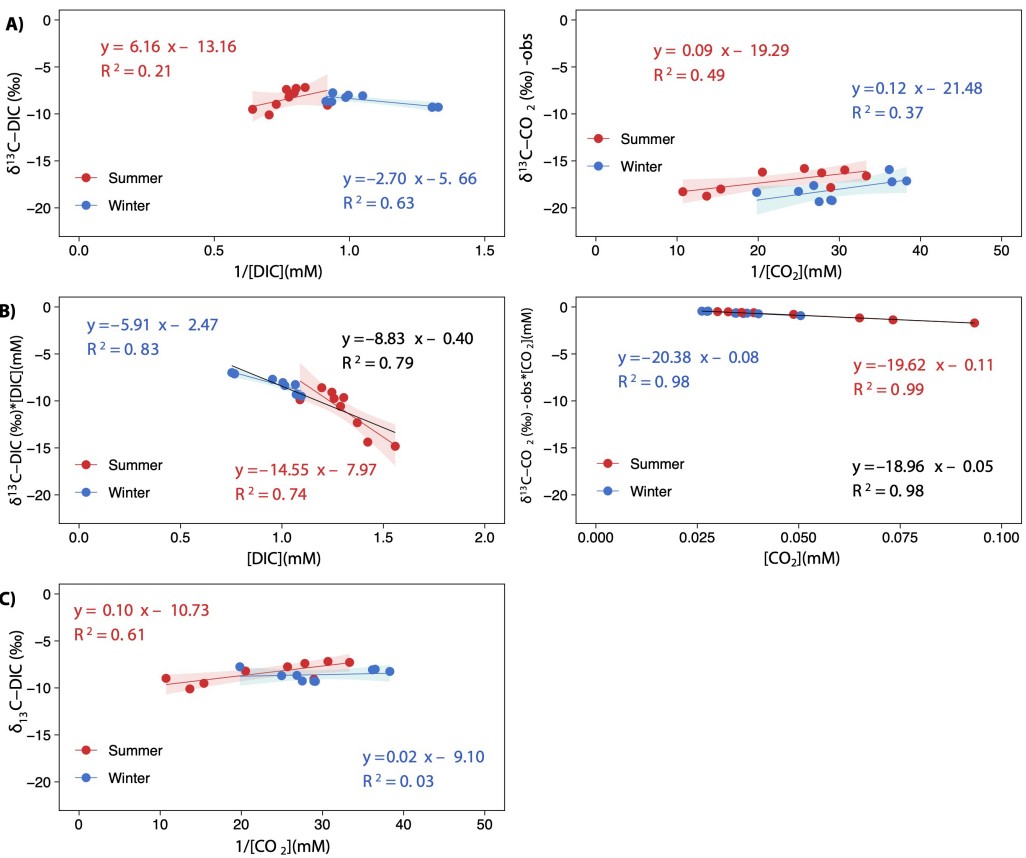

**Figure 7** **Keeling plots (A) and Miller-Tans plots (B) analyses for $\delta^{13}$C-DIC and $\delta^{13}$C-CO$_2$ observed values.** A scatterplot showing the relationship between $\delta^{13}$C-DIC and the inverted CO$_2$ concentration [1/CO$_2$]. The points are colored according to the season (summer and winter), and the regression lines are plotted for both seasons and the whole annual data (equation in black), when statistically significant. All regressions were significant for $p < 0.05$, except for winter in (C) which was not significant ($p = 0.65$). The shaded areas represent the 95% confidence level of each linear regression.

the observed lake metabolic activity in our lakes. Moreover, the absence of a statistically significant relationship between NEP, C$_{CO2,}$ and $F_{CO2}$ suggests that other factors more important than the metabolic processes control the dynamics of CO$_2$ in these carbonate lakes. Our results support the strong phosphorus limitation for primary production in Ruidera Lakes, which are therefore not capable of modifying the CO$_2$ dynamics. The significantly high phosphorus concentration observed in winter did not stimulate primary production because this was prevented by the low water temperature. When lake waters become warmer in summer, SRP is no longer available because it is consumed in another abiotic process. The latter is crucial as it implies a more significant phosphorus availability effect than water temperature on controlling the CO$_2$ sink capacity in oligotrophic lakes. On the other hand, the positive relationship found between the concentration of C$_{CO2}$ and N-NO$_3$ is inherent to the primary source of both through the groundwater discharge into the lakes. The use of fertilizers for agricultural purposes in the Campo de Montiel aquifer
resulted in a significant increase in nitrate concentration in the groundwater (*i.e., Lima, 2008*; *Soriano & Álvarez, 2016*).

The dissolution of carbonate rocks leads to the production of DIC and to the release of $Ca^{2+}$ ions into the karst waters (*Hamilton et al., 2007*), a feature evidenced in Ruidera Lakes (*Alvarez-Cobelas et al., 2006*). Our results suggest that DIC inputs into the lakes act as a driver of $C_{CO2}$ dynamics. The signatures of $\delta^{13}$C-DIC recorded in all our lakes during both seasons below $-12$‰ confirmed the primary geogenic influence on DIC, as reported by *Deines, Langmuir & Harmon (1974)* for other karst lakes. The headwater position of the lakes within the watershed, the carbonate lithology, the dominant groundwater discharge, and the limitation of phytoplankton metabolism mean that the abiotic components of the DIC regulate the $CO_2$ dynamics in the lakes. However, the Miller-Tans regression models also suggested $\delta^{13}$C-DIC seasonal differences, reflecting the influence of other processes during summer months on DIC dynamics. Once the biotic effect has been ruled out, other abiotic processes must make the $\delta^{13}$C-DIC signal more negative. As suggested by the calcite saturation index, carbonate mineral precipitation could also be driving the excess of $C_{CO2}$ and $F_{CO2}$ in Ruidera Lakes, as stated by *Cao et al. (2023)* in their recent review on $CO_2$ in karst aquatic systems. The computed signature of $\delta^{13}$C-$CO_2$ released from calcite precipitation ($\delta^{13}$C-$CO_2$-fractionation) in the three studied lakes strongly supports the importance of this abiotic process in the $CO_2$ dynamics. Calcite precipitation leads to a pH decrease, which promotes $CO_2$ outgassing by affecting the carbonate equilibrium, as revealed by the negative relationship between $CO_2$ and pH. Miller-Tans plot disclosed that the $CO_2$ source in the lakes is close to this $\delta^{13}$C-$CO_2$-fractionation and the main contribution to the $C_{CO2}$ should be attributed to the calcite precipitation process, supporting the key effect of temperature on the solubility of calcium carbonate, according to the seasonal change observed in $\delta^{13}$C-DIC values.

Terrestrial carbonate weathering is a temporary sink for $CO_2$ which is partially reversed by calcite precipitation but then counteracted by biological uptake (*Nõges et al., 2016*; *Perga et al., 2016*). The increase in phytoplankton productivity should buffer this excess of $CO_2$ released from the abiotic origin, but phosphorus is limiting the C sink behavior of these lakes. Co-precipitation of phosphate with calcite may also be a cause of phosphorus limitation on algal growth in carbonate-rich lakes (*Hamilton et al., 2009*). This phenomenon has been observed in the recent sedimentary tufa deposits of one of the lakes in Ruidera (Tomilla lake; *Souza-Egipsy et al., 2006*); the authors highlighted this mechanism of calcium phosphate co-precipitation acting in the first stages of calcite mineralization, but unrelated to the biogenic origin of calcite precipitation. The joint diel variation SRP and T could be supported by phosphate co-precipitation. The lower concentration of SRP in summer would also be explained by temperature-dependent co-precipitation with calcite. Calcite precipitation and the consequent phosphate removal might act as a negative feedback to eutrophication. However, the net environmental benefits generated by this abiotic process of eutrophication are mitigated by increasing greenhouse gas emissions from lakes. The seminal article by *House (1987)* warns that as eutrophication increases and the SRP content increases, the co-precipitation of phosphate with calcite becomes less effective because the SRP progressively inhibits the precipitation reaction. Thresholds were

already noted experimentally years ago (*House, 1990*). *Plant & House (2002)* suggest that under concentrations of SRP above 20 $\mu$mol L$^{-1}$, calcite precipitation is inhibited, which is expected to occur in meso and eutrophic systems. However, it is not well known how the presence of $CO_3^{2-}$ ions controls the co-precipitation dynamics in lakes. It is necessary to establish how they could interact by changing the magnitude of the mentioned processes and $CO_2$ emissions. In groundwaters, for example, reactions that remove these calcite nucleating inhibitors (SRP and DOC) have been identified. However, they disappear when calcite saturation is maximized under low-flow conditions in lake waters (*Neal, 2001*), but the main causes are yet unknown.

According to our results, the observed variations in $F_{CO2}$ between studied lakes suggest that urbanization did not foster $C_{CO2}$ and $F_{CO2}$. Nonetheless, the scale of urbanization in Ruidera Lakes is still small compared to the large scale of other natural abiotic processes such as calcite precipitation. In short, DIC is provided throughout the year by the carbonate weathering in the karst aquifer discharges at the upstream lake LC, encouraging calcite precipitation reaction and the subsequent $CO_2$ surplus in the water column, which is efficiently released into the atmosphere. This temperature-dependent reaction is enhanced in summer, increasing $CO_2$ emissions and massively evacuating $CO_2$ from the water column. Downstream, $CO_2$ emissions in lakes decreased, but contrary to the expected, $\delta^{13}$C-DIC signature was enriched due to the effect of calcite precipitation. The amount of $CO_2$ released by the heterotrophy is negligible compared to calcite precipitation. Overall, during the summer, a large amount of $CO_2$ evasion into the atmosphere was confirmed in this study, indicating that the Ruidera Lakes are a source of carbon with an average seasonal release.

## CONCLUSIONS

Lakes and other freshwaters release almost as much $CO_2$ as all the Earth's oceans. A large amount of allochthonous $CO_2$ is discharged into the lakes from both terrestrial biomes and geological sources. However, it is usually buffered through plankton metabolisms, lowering $CO_2$ emissions and sequestering C in the sediments. However, all lake types follow a different pattern because other processes determine the carbon budget. In carbonate-rich karst lakes, the abiotic processes play an essential role in the $CO_2$ balance. Bicarbonate-rich groundwaters entering karstic lakes should promote algal growth, but P affinity for abiotic processes drives the carbon budget. Calcite–phosphorus co-precipitation impinges as a negative feedback to lake eutrophication, promoting consistent $CO_2$ release from karstic lakes. In addition, the P-limitation of phytoplankton—and probably of other microbial processes—cannot exert any control over the excess $CO_2$ that is released during the consistent precipitation of calcite. Geochemical processes such as calcite precipitation act as the main driver of $CO_2$ in carbonate-rich lakes, providing an excess of $CO_2$ that is consistently released into the atmosphere. However, in highly eutrophicated carbonate lakes, the co-precipitation of SRP with calcite seems to be inhibited and algae is no longer P-limited, but $CO_2$ emissions continue to be high. This indicates the existence of certain thresholds that must be determined to understand how abiotic (calcite precipitation) and

biotic (heterotrophy) processes are interconnected with each other to control the excess $CO_2$ generated. This is important in the global C budget because although carbonate regions extend over 15% of the Earth's surface, $CO_2$ dynamics are in this case likely subject to dynamics as intense as those emerging from volcanic areas.

## ACKNOWLEDGEMENTS

Thanks to the Staff of Lagunas de Ruidera Natural Park and Matías Hotel for their support during the field campaigns. We also appreciate the enormous effort of our colleagues Andrea Guzmán and Ismael Soria (UNAM) and José Luis Ayala (MNCN-CSIC) for their support in the fieldwork and laboratory analyses.

### Funding

This study was supported by the Grant PID2020-116147GB-C21/AEI/ 10.13039/501100011033 (DAMOLAKE) funded by the Spanish Ministry of Science and Innovation (MCIN/AEI) through the "European Union NextGenerationEU/PRTR" supported this research. Programa de Posgrado en Ciencias del Mar y Limnología (UNAM) and CONAHCYT provide a doctoral scholarship to MVS (CVU: 828722). Additionally, COOPA20433 funded by CSIC within the i-COOP+2020 Program, also supported this research. The funders had no role in study design, data collection and analysis, decision to publish, or preparation of the manuscript.

### Grant Disclosures

The following grant information was disclosed by the authors:
Spanish Ministry of Science and Innovation (MCIN/AEI): PID2020-116147GB-C21/AEI/10.13039/501100011033.
Programa de Posgrado en Ciencias del Mar y Limnología (UNAM).
CONAHCYT provide a doctoral scholarship to MVS: CVU 828722.
CSIC: COOPA20433.

### Competing Interests

The authors declare there are no competing interests.

### Author Contributions

- Mariana Vargas-Sánchez performed the experiments, analyzed the data, prepared figures and/or tables, authored or reviewed drafts of the article, and approved the final draft.
- Javier Alcocer analyzed the data, authored or reviewed drafts of the article, and approved the final draft.
- Eric Puche analyzed the data, authored or reviewed drafts of the article, and approved the final draft.
- Salvador Sánchez-Carrillo conceived and designed the experiments, performed the experiments, analyzed the data, prepared figures and/or tables, authored or reviewed drafts of the article, and approved the final draft.

## Data Availability

The data is available at DIGITAL.CSIC: http://hdl.handle.net/10261/340351.

## Supplemental Information

Supplemental information for this article can be found online at http://dx.doi.org/10.7717/peerj.17393#supplemental-information.

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
