# Peer review of "Abiotic processes control carbon dioxide dynamics in temperate karst lakes"

_PeerJ, doi:10.7717/peerj.17393_

## Round 0.1 · original submission · Major Revisions

Based on the comments of the reviewers and my assessment, there are several improvements are needed before the manuscript is accepted for publication. You need to improve the introduction by making it more focused, for instance by focusing more on mechanisms and drivers of CO2 emission and not that of methane. You also need to improve the statistical analysis and interpretations as suggested by Reviewer 1. You also need to justify the use of diffusive gas fluxes using static chambers and any assumptions made in that regard.

Reviewer 1 ·

Basic reporting

No comment

Experimental design

No comment

Validity of the findings

No comment

Additional comments

This paper looks at the role of abiotic processes, such as calcite precipitation, in controlling carbon
dioxide dynamics in 3 temperate karst lakes during the winter and summer. Overall, the introduction is well-written, and the experimental design is valid. However, some key issues still need to be solved before publishing the manuscript. These are highlighted in three crucial points, with further details embedded in the General comments.

Materials and methods: -
While the statistics are well described, some information is still missing. The primary analysis of the work looks at four factors influencing the independent variables. These factors are lakes, depth, seasonality, and time of day. In the current description, the depth factor is not included but appears in the results. Post hoc test results are also missing, meaning it's difficult to tell, for example, which lake or depth led to the overall significant influence of the factor. I also think the PCA analysis was added but not utilized well in analyzing the controls. While having different statistical methods suits the paper, it can be a bit overwhelming to the reader. Consider linking the PCA results to the MRM or removing the former entirely.

Results:-
While the results were mainly well described, some structural adj ustments are needed to
make the section more readable. For example, the authors are looking at differences in their datasets related to 4 factors (Lakes, seasonality, depth, time of day). At the moment, everything is mixed up; thus, the results section is difficult to follow. I suggest starting with the time section, i.e., seasons and time of day, and then moving to the space section, i.e., lakes and depth, in the description. That will help the reader alot in following. Also, the statistical analysis of significance would be better placed in a table where all the independent variables are included, and then the text section can be a bit less crowded. The text also describes the results, implying that the interaction between season and lake was also considered in the statistics, which I believe was not. The authors need to clarify whether they looked at individual factors (season+time of day+depth+lake) or their interaction (season*lake…..) in their models.
Grammar mistakes: -
The English language should be improved to ensure that an international audience can clearly understand your text. These include missing words, hanging sentences, wrong words, and the misplaced use of the article “the.”

Graphics:-
Captions need to be rechecked as they miss important information. The graphics also
need to be improved, i.e., declutter the y-axis labels.

General comments
Main text
Introduction
Ln 124-128: Consider moving this to the discussion/conclusion section as an outlook/hypothesis, as it seems misplaced here. The scope of the study does not allow an analysis of the effects of global change on Karst Lakes' CO2 evasion rates as it is only limited to 3 lakes.
Materials and methods
Ln 160: Consider changing to sampling campaigns. The current word does not exist . Check here and other places in the manuscript.
Ln 177: Does the 13mm represent diameter or pore space size? Consider clarifying and adding the pore size, too, as it defines what is considered to be POC and DOC.
Ln 517: Check subscript here and line 520. I expected the enrichment to increase as CO2 is evaded, i.e., loss of the lighter inorganic carbon as CO2.
Conclusion
Ln 538: Missing word
Ln 539-540: This contradicts your earlier hypothesis, which was that in high p environments, primary production will be favored, the calcite precipitation inhibited, and more CO2 will be stored due to the increased production

Figures and tables
Table 2: Add post hoc results in the form of different letters here.
Table 3: Add in the caption that beta refers to the slope of the relationship. I suggest only having the final R2, as the results section did not discuss the change.
Table 4: Post-hoc test results also required
Figure 1: How are the satellite maps showing urbanization influence? Consider clarifying this in the figure and the text.
Figure 2: Correct to Boxplot. Consider adding the Post-hoc letters in the figures to show differences within grouping factors, i.e., season and lake. The seasonal notation is missing.
Figure 3: Same comments as above.
Figure 5. Figure caption and figure do not match. What's the equation in black showing?
Figure 7B: What's the equation in black showing? Information missing in the caption

Annotated reviews are not available for download in order to protect the identity of reviewers who chose to remain anonymous.

Reviewer 2 ·

Basic reporting

This manuscript describes a 2-season campaign monitoring carbon dioxide (CO2) concentrations and fluxes in three karstic ecosystems, nevertheless the introduction is unclear about some topics, e.g., Importance of karstic water bodies in global context and some processes involved in CO2 cycle.
I am confused as to why introduce some processes related to CH4, as methanotrophy and methanogenesis if they will not be described or treated in the development of the manuscript.
Line 57. The mention of CH4 is a bit confusing, why mention specific processes that will not be discussed in the development of the article.
Line 79. Put the water bodies in karst areas in global context, are the important in global context?
Line 96. Which other microbial processes compete for C, specify please.
Line 110. Why abiotic processes are usually exacerbated in karst lakes?
Line 127. It is outside the focus of this work; “sound and effective environmental management practices to mitigate climate change” are not proposed.

Check orthography, e.g.: Line 465 and Line 520

Experimental design

It is necessary to justify the experimental strategy, mainly consider the problems associated with quantifying diffusive gas fluxes using static chamber including those mentioned by Mattews et al., 2003 (DOI:10.1021/es0205838).
Line 202. With 30-45 min of monitoring is necessary to consider possible overpressure and its alterations on the air-water equilibrium. Is the behavior of CO2 concentration in the headspace of the chamber linear?

Validity of the findings

It is necessary to list the assumptions considered to validate the scaling of the measurements in this work.
Line 363. It is not necessary to mention a poor correlation like R2= 0.16 unless that correlation implies something important, in which case it would be good to show it graphically.
Line 388. Considering that this MS describes a 2-season campaign monitoring CO2 concentrations and fluxes in three karstic ecosystems, is the sampling strategy sufficiently robust to upscale and describe the annual lakes behavior? Argue.

---

## Round 0.2 · Minor Revisions

I invite you to respond to the remaining minor comments raised by the reviewer before I make further recommendations for your manuscript.

Reviewer 2 ·

Basic reporting

The current manuscript has taken into account the corrections and suggestions of the reviewers. However, I still have some doubts regarding the experimental procedure, which I will state in later sections, which I hope will be answered, after which I recommend publication of this manuscript in your journal.

Experimental design

Line 213.- The overpressure in the assays is mainly due to the fact that the emissions are high, so that the concentration in the headspace modifies the equilibrium between the air-water interface. Rather than removing, I consider it necessary to guarantee linearity in the behavior by regulating the tests over time or to know the error caused by this overpressure and correct it.
However, in the case of removing some data, specify how many.

Validity of the findings

no comment

---

## Round 0.3 · accepted · Accept

Following the changes made by the authors following the reviewers' comments and my own review, I deem, the revised manuscript acceptable for publication in its current status.